# “I Just Want Some Clear Answers”: Challenges and Tactics Adopted by Migrants in Denmark When Accessing Health Risk Information about COVID-19

**DOI:** 10.3390/ijerph18178932

**Published:** 2021-08-25

**Authors:** Rasmus Luca Lyager Brønholt, Nina Langer Primdahl, Anja M. B. Jensen, An Verelst, Ilse Derluyn, Morten Skovdal

**Affiliations:** 1Department of Public Health, University of Copenhagen, Øster Farimagsgade 5B, 1014 Copenhagen, Denmark; nina.primdahl@sund.ku.dk (N.L.P.); anja.jensen@sund.ku.dk (A.M.B.J.); m.skovdal@sund.ku.dk (M.S.); 2Centre for the Social Study of Migration and Refugees, Department of Social Work and Social Pedagogy, Ghent University, H. Dunantlaan 2, 9000 Gent, Belgium; An.verelst@ugent.be (A.V.); ilse.derluyn@ugent.be (I.D.)

**Keywords:** health risk communication, COVID-19, migrant health, social media

## Abstract

Health risk communication plays a crucial role in preventing the spread of infectious disease outbreaks such as the current coronavirus (SARS-CoV-2). Yet, migrants are far too often forgotten in health risk communication responses. We investigate the challenges and efforts made by migrants in Denmark—in the initial months of the pandemic—to access information about COVID-19. We draw on 18 semi-structured interviews conducted in May and June 2020. All interviews are thematically coded and analyzed. Our analysis reveals that many of the migrants faced several challenges, including accessing information in a language understandable to them and navigating constant streams of official news flows issuing instructions about which actions to take. However, we also note that the participating migrants found numerous creative ways to address some of these challenges, often aided by digital tools, helping them access crucial health and risk information. This paper highlights that migrants constitute an underserved group in times of crises. They are vulnerable to getting left behind in pandemic communication responses. However, we also identify key protective factors, social resources, and agentic capabilities, which help them cope with health and risk information deficits. National governments need to take heed of these findings to inform future pandemic responses.

## 1. Introduction

Health and risk communication are critical to prevent the spread of the severe, acute respiratory syndrome coronavirus 2 (SARS-CoV-2), Health and risk communication are critical to prevent the spread of the severe, acute respiratory syndrome coronavirus 2 (SARS-CoV-2), also when it comes to the roll out and effectiveness of vaccines, yet it brings with it challenges and complexity [1]. Only when individuals have an accurate understanding of their risk of getting infected by or dying from the coronavirus disease 2019 (COVID-19) will they be able to take responsibility and engage with the range of prevention practices and technologies available [1]. Furthermore, studies have also shown that some migrants have an increased risk of contracting infection with COVID-19 as well as suffering severe consequences from it [2,3,4], including psychosocially (e.g., depression, anxiety, or stigma) [5]. It is therefore essential that information about COVID-19 is made accessible to all population groups, including migrants, giving them an equal chance to find effective ways to protect themselves from infection and engage with guidelines concerning preventive measures in their country of residence [6,7,8,9]. However, migrants are often “left behind” in health risk communication efforts [6,10]. A review of public health communication during the first months of the COVID-19 pandemic revealed that fewer than half of European countries had online COVID-19 advice in at least one major migrant language [10]. In Denmark, official information about COVID-19 (e.g., how to avoid infection or information on testing) in English was not published until the end of March 2020, approximately two weeks after the first national lockdown [11]. Information about COVID-19 has since been made available in other languages [12,13]. Other sources of news also failed to consider other languages. For example, live press conferences frequently held by the prime minister and health authorities were broadcasted without subtitles or translation [13]. As a consequence, and as noted by the World Health Organization ‘ApartTogether’ survey, many refugees and migrants report relying on social media or news about COVID-19 from their country of origin to keep themselves informed [14].

Failure to consider migrants’ linguistic diversity in health risk communication responses leaves them underserved and vulnerable [15] and ultimately put migrants at higher risk of SARS-CoV-2 transmission [3]. For example, a report from the Danish Institute for Human Rights interviewing ethnic minorities with poor Danish language skills during the COVID-19 period showed that some were uncertain about the risk of transmission of the disease as well as the authorities’ guidelines, which sometimes led to extreme self-isolation and concern [16]. A small but expanding body of literature is beginning to highlight the consequences of the linguistic challenges faced by migrants in this time of crisis. For instance, research from Nepal [17] and the Netherlands [18] has found language barriers to be associated with numerous consequences such as stress, anxiety, difficulties in accessing relevant services, and the spread of misinformation among different migrant groups [17,18,19]. Furthermore, not addressing migrants’ linguistic diversity does not just leave migrants underserved and vulnerable to infectious diseases such as COVID-19, it undermines the broader pandemic response.

Much current knowledge, however, is based on quantitative data, commentaries, or editorials. We have little empirical understanding of how migrant groups qualitatively experience their access (or otherwise) to health and risk information concerning COVID-19. Furthermore, much of what we do know relates to the information barriers migrant groups face in accessing health and risk communication, whilst little has been done to explore the tactics they adopt to overcome these barriers. We therefore take inspiration from de Certeau’s [20] notion of ‘tactics’ to explore the everyday acts and occurrences that migrant groups adapt to attain some control, or make do, in an information and communication setting deemed constraining. By drawing on the concept of ‘tactics’, we will be able to not only highlight the challenges experienced by our participants but also disentangle their actions and the supportive structures they form part of. Against this background, we ask: what are the tactics adopted by migrants in Denmark when facing the challenges of accessing information on COVID-19?

## 2. Materials and Methods

We drew on data from a qualitative interview study that aimed to investigate the various impacts of the COVID-19 pandemic for migrants living in Denmark. The study forms part of the international ‘ApartTogether Study’, which is a collaboration between WHO, the UN System, and a consortium of universities (coordinated by the University of Copenhagen and Ghent University) [13]. Informed oral consent was obtained from all participants, guaranteeing that they would be anonymized. We used pseudonyms when referring to the participants throughout this paper.

### 2.1. Study Location and Participants

The study took place in Denmark during the spring of 2020, following the first wave of the COVID-19 pandemic. Like many other countries, the Danish government implemented a nationwide lockdown of all non-essential activities. However, at the time of the interviews, most activities had temporarily reopened [20].

Recruitment of the migrants used in this study took place in May and June of 2020. Migrants who had filled out the ApartTogether Study survey and who had indicated that they would be willing to participate in an interview were contacted. Only respondents who had answered the survey in English or Danish were contacted. Interviewees were contacted through either e-mail or a text message depending on the contact information provided. After a few days, a reminder was sent to interviewees who had not yet answered our first request. We recruited migrants of any age, interviewing a total of 18 migrants living in Denmark. The migrants included four men and 14 women. They came from various countries within and outside of Europe. Many of the migrants had lived in Denmark for a year or less, and there was a range between four months and seven years for all migrants (see Table 1).

### 2.2. Data Collection and Analysis

In May and June 2020, the first author conducted 18 semi-structured interviews with the migrants. Due to the COVID-19 restrictions imposed by the Danish government at the time of the interviews as well as the general risk of gathering, all interviews were conducted virtually (on Zoom or by phone). A description of the study was provided to the participants followed by obtaining an oral consent. All interviews were conducted in English and lasted between 31 and 76 min. The interviews were steered by a topic guide structured around five different themes: (1) corona and social distancing in your everyday life, (2) experiences and sense-making, (3) worries and vulnerabilities, (4) social support and coping strategies, and (5) the response of the authorities and civil society. The conversations surrounding health risk communication are the center of this article.

All 18 interviews were audio-recorded and subsequently transcribed in their full length. In our efforts to disentangle emerging themes, we coded and thematically organized the interview transcripts in accordance with the steps outlined by Attride-Stirling [21]. This was done with the support of NVivo 12, a qualitative data analysis software. A total of 112 codes emerged from this process. However, we do not intend to present all the emerging themes in this paper. Instead, we focus on the 28 codes that relate to health risk communication challenges and tactics (see Appendix A). The remaining codes covered themes such as worries about their financial situation, well-being of their relatives, and loneliness. In preparation for this paper, the 28 selected codes underwent a second round of organizing. Codes that related to each other were grouped into basic themes. If the basic themes shared similar issues, they were then rearranged into what Attride-Stirling refers to as more interpretive organizing themes (see Table 2). In the following analysis, we will present and unpack the different themes derived from our thematic network analysis.

## 3. Results

Throughout the interview narratives, it was clear that many of the migrants faced several challenges when it came to accessing the health risk communication coming from the Danish authorities. In response to this, however, many of them found other channels of communication.

### 3.1. Challenges: “I Just Want Some Clear Answers”

Throughout the interviews, many of the migrants expressed that it was difficult to navigate the authority’s response to COVID-19. This included how and where to find health and risk information, or making sense of the information they found, either because it was in Danish or imbued with cultural reference points. For instance, one common issue among the migrants was not knowing which and when restrictions were put in place. This is illustrated by Leah, who had been living in Denmark for only nine months. In the following quote, she describes her inability to follow the regularly press briefings held by the Danish authorities in the spring of 2020 at the height of the first wave of the pandemic:

“It has been really hard to figure out like when Danes know [e.g., timing of press briefings]. Like that the prime minister is going to make a speech. Because it will happen at seemingly completely random times. And then like all the sudden we find out “oh yeah, she has been speaking for half an hour” and we turn it on. But like there was no way to figure out. Yeah, I never, I still don’t know how people are finding out when she was going to say important things. I was always like a bit behind”.(Leah, age 33)

Not knowing what was going on made it difficult for her to stay updated on the rapidly changing measures imposed by the Danish authorities:

“It’s still kind of difficult to figure out how to navigate certain things. Like what things are open and, you know, we didn’t know that you could get like a Coronavirus test until someone told us it. Like we haven’t gotten any communication from the Danish government at all. About what we should be doing or anything like that”.(Leah, age 33)

The feeling of uncertainty about what measures applied was also highlighted by Heather who had lived in Denmark for less than a year. When asked about whether she needed more support to cope with the pandemic, she reflected on the communication efforts towards migrants coming from the Danish authorities:

“But I don’t think I need more support exactly; I would just want some clear answers on what is allowed and what’s not. (…) It’s probably because we are not counted as well, but there is a lot of things that is about Danish citizens and not about foreigners”.(Heather, age 31)

These accounts can be read as examples of recently arrived migrants feeling left out of the response due to their migrant status and lack of proficiency in the Danish language. Besides being a great source of frustration among several of the migrants, this sense of not being “counted”, as Heather puts it, resulted in confusion and uncertainty about how best to navigate the pandemic response (e.g., Leah not knowing she could get a COVID-19 test or Heather not knowing what was “allowed”).

Another common barrier to the migrants’ ability to navigate the pandemic response was linguistic difficulties. Many of the migrants did not speak Danish, making it difficult for them to understand much of the COVID-19-related information coming from the Danish authorities. Matej, who had just started learning Danish, provided an example of his inability to follow the Danish news:

“So most of the time I feel like that most of the information is in Danish, and I started to learn Danish recently, but I am not fluent yet”.(Matej, age 25)

Faiza, who had been in Denmark for less than a year and did not speak any Danish yet, also described how information from the Danish authorities are not accessible for migrants:

“Information is not that clear. I mean, it is not eligible for people who are like us, immigrants”.(Faiza, 35)

Despite Faiza knowing how to navigate the news flows, she was still feeling left behind due to linguistic barriers. Many of the migrants interviewed did not feel that they were addressed in a way that was understandable to them, and this was especially true for those migrants who had not been in Denmark for very long or had moved to Denmark alone. Because of their unfamiliarity with the Danish system, coupled with their inability to understand the Danish language, many of our participants were unable to understand the messages and health advice coming from official sources, e.g., the regular press briefings. Like Faiza, they were lost in translation.

Shortcomings by the Danish state to make health risk information available to migrants in a timely manner were, however, partly addressed by other structures, such as workplaces or language schools. This was experienced by a small handful of our participants. For instance, Irene, who had come to Denmark only a year ago, explained that the language school that she was enrolled in not only provided Danish lessons but also valuable COVID-19-related information in the beginning of the pandemic:

“When it started to sort of get momentum, the whole pandemic. (…) In school we had a class that was completely dedicated to, you know, “this corona is looming, wash your hands”, you know, “sneeze in your elbows”. So like this sort of, they basically also explained the basic things about, you know, personal hygiene and safety.”(Irene, age 37)

For others, the workplace was a way to stay updated on what was going on with regards to changing COVID-19 rules and regulations. This was exemplified by Bogdan, who regularly received essential information about COVID-19 from his manager:

“Our general manager would send us e-mails. Keeping us posted with what’s going on. (…). Most of the time actually all the time, we knew what’s going on. And they would tell us, you know: “when we gonna know more from the authorities, we’re gonna let you know””.(Bogdan, age 38)

However, for most of our participants, such ‘back-up’ support was unavailable, requiring them to actively seek out information.

### 3.2. Tactics: Seeking to ‘Be in the Know’

To get information about COVID-19 in Denmark, the migrants employed various tactics. Tactics can be understood as the formation of new daily practices amongst people in marginalized positions as a result of their absence in more formal, dominant and strategic responses, either to ‘get by’ or to display resistance [19]. In this case, the marginalized positions are those of migrants in Denmark, who experience exclusion from the authorities’ channels of communication. They therefore turn towards opportunities to receive information about COVID-19 provided outside the established frameworks of communication.

One tactic was to use a tool to translate, for example, press briefings held by the authorities, here exemplified by Linh:

“And everyone was just watching the news, and again I don’t speak any Danish, but I just used the translator on my web browser to try to see if, what the authorities say, at least when they are on TV or the newspaper”.(Linh, age 30)

Other tactics that the migrants employed on their own included reading online news in English or their native language or accessing news from their country of origin. Beyond searching for information on their own, some migrants also relied on their close social networks such as family members or friends who spoke Danish. Faiza, who had a Danish boyfriend, described that he translated the most important news about COVID-19:

“So, it’s difficult and my boyfriend is happy to translate it. If relevant he translates it to me, but then I would like to process it on my own, you know”.(Faiza, 35)

Some migrants also used their broader social network to stay updated with the aid of social media. Here exemplified by Blanca, who was a part of a WhatsApp group with her friends (from the same country as Blanca) also living in Denmark at the time. For example, they regularly shared COVID-19 related information with each other:

“When it comes to information from Denmark, we, what we had to do, or didn’t do, we, we had a WhatsApp-group with our friends here in Copenhagen, and some of them have been here for many years, so they would send links with important information, and every time that Mette [the Danish Prime Minister] talked, they would just send a WhatsApp “so hey, that is kind of like the summary”, so they kind of like made it work for us”.(Blanca, age 35, Spain)

As the accounts show, Faiza and Blanca made active use of their social networks to access and share information. However, even though both were able to mobilize their social network to get relevant information regarding COVID-19, they also expressed a desire to assess the information on their own without relying on the help of others or digital tools.

In our interviews, information sharing about COVID-19 also happened in much more structured transnational networks. Several of the interviewed migrants explained how they actively used larger groups or communities on social media platforms that targeted migrants living in Denmark. This was well articulated by Ana, who did not speak any Danish and did not have any Danish-speaking relatives and friends to help her with translation of COVID-19-related information. In the following, she reflects on how she stayed updated using Facebook:

“(…) So it is a Facebook group. And there are some people there that right after the government announcements, they translate everything into bullet points what is most important, so mainly I would follow all the guidelines from there”.(Ana, age 34)

Ana continues, elaborating on how the group supported her in accessing COVID-19-related news:

“The Facebook group actually helps quite a lot, because there are some people that they have been through the struggle of the language, and they know how difficult it is to actually have access to news etc. So you have a lot of support of people that have been living on the country, ehh, for many, many years, and they are actually there to support you and give you information”.(Ana, age 34)

Having just arrived in Denmark, Ana has not had time to adapt to her new surroundings. Therefore, the Facebook group was particularly important for her to stay updated on the COVID-19 response in Denmark. For newly arrived migrants who still needed to establish strong social networks, the group constituted an important access point for COVID-19-related information, including knowledge about measures introduced by the Danish authorities, and ultimately being able to follow guidelines as they relate to the Danish context. The use of such social media networks can be seen as an example of how the sudden reorganization of established spaces, because of COVID-19, was tactically utilized by migrants—providing them with important information about COVID-19.

## 4. Discussion

We set out to explore the challenges and tactics adopted by migrants when accessing COVID-19 information. Our findings show that being away from their country of origin in a time of crisis, and having no or only limited knowledge of the Danish system and language, some migrants find themselves excluded from the COVID-19 response. Most of the migrants participating in our study noted that much of the information available at the start of the pandemic was in Danish and that it was difficult to navigate the news flows. Having little knowledge about the Danish language and customs, many of the migrants were unable to keep up with the changing restrictions and guidelines and the rationale behind these. Our data suggest that this was especially the case for migrants who had only been in Denmark for a short period of time and who had come here alone. Those migrants who had stayed in Denmark for several years appeared to face fewer challenges in navigating health risk information about COVID-19. The feeling of not being included in the response—or repeatedly accessing information with much delay—led to uncertainty and frustration among many of our participants. Whilst the Danish authorities and non-governmental organizations eventually made essential health and risk information available in multiple languages, it was difficult for the participants to keep up with changes, often announced in press conferences held in Danish. Our participants therefore relied on adopting various tactics to access and share information. These included the use of translation tools, searching for news in their country of origin and on social media platforms, as well as engaging with different digital migrant-networks, helping each other navigate the pandemic in Denmark.

In different ways, digital tools emerged as critical enablers of tactics in accessing and sharing health and risk information outside of established structures and spaces. This finding resonates with previous observations. A study by Park et al. [22], exploring the experiences of migrants living in the Republic of Korea at the time of the MERS outbreak, describe digital tools as “weapons to survive” an emergency when living in a foreign country. They found that digital tools were used to receive and share information among migrants in their closer social networks [22]. In our study, however, information sharing also happened in much larger online social media groups and communities, illustrating new tactical practices. Social media thus acted as a key gateway to receive health and risk information about COVID-19 when official information channels were inaccessible. 

Despite social media platforms being a much-used alternative source of information on COVID-19 among the migrants in this paper, studies suggest that social media platforms also have the potential to disseminate inaccurate information [22,23]. Furthermore, it has been suggested that increased use of digital tools to get COVID-19-related information, and the use of many diverse platforms, can lead to information overload, increasing the risk of important information to be ignored or forgotten [24].

While digital tools were important for the migrants in accessing health and risk information about COVID-19, the tactical use of such tools often also involved fellow migrants or other people in general. Even though digital tools enable access to information, these digital tools are only relevant because the migrants we spoke to were part of compassionate social networks characterized by information sharing and support. Relying solely on the help of others (and often strangers), however, to access vital health information might leave them vulnerable to disruptions in information flows or misinformation. Nonetheless, given the absence of the Danish welfare state to cater for the health and risk information needs of migrant populations in the early and critical weeks of the pandemic, digital tools proved critical in helping our participants to access essential health and risk information when there was no alternative.

Our findings are constrained by two methodological limitations. One, we did not have access to language interpretation. This meant that we could only recruit participants who either spoke English or Danish. As many of the participants did not have English or Danish as their first language, important nuances of their experiences may have been lost. Two, we recruited participants and conducted the interviews through online and digital means, requiring a certain level of digital literacy and access to digital tools. It is therefore likely that we have recruited more resourceful migrants, missing out on the experiences of those who are most vulnerable. Future research ought to include more disenfranchised migrant populations.

Whilst our findings highlight the practical implications of social media platforms in helping our (more resourceful) participants to access information about COVID-19, they also point to the need for offline alternatives to reach more vulnerable migrant groups. Council estates with significant migrant populations could recruit trusted COVID-19 ambassadors for different cultural language groups and support them go door-to-door making COVID-19 health risk, testing, and vaccine information available.

## 5. Conclusions

This paper demonstrates the agency and resourcefulness of migrants in overcoming the specific challenges some migrants face when it comes to their ability to access and understand health risk communication in the country they reside in. Understanding and recognizing the role migrants can play in health risk communication responses is key to support them during this pandemic and in the future. There is a need to consider migrants’ resources when developing future information strategies, including involving and engaging migrants in the process.

## Figures and Tables

**Table 1 ijerph-18-08932-t001:** Participant characteristics.

Name	Gender	Country of Origin	Age	Time in Denmark	Migrant Status	Date of Interview
Ana	Female	Latin America	34	4 months	Temporary documents	9 June 2020
Julia	Female	Eastern Europe	28	7 years	Temporary documents	4 June 2020
Viviana	Female	Latin America	33	6 months	Temporary documents	3 June 2020
Faiza	Female	Middle East	35	10 months	Procedural stay ^1^	3 June 2020
Matej	Male	Eastern Europe	25	1 year, 10 months	Permanent documents	5 June 2020
Leah	Female	North America	33	9 months	Temporary documents	3 June 2020
Gosia	Female	Eastern Europe	30	10 months	Permanent documents	10 June 2020
Heather	Female	Western Pacific	31	10 months	Temporary documents	11 June 2020
Karim	Male	Middle East	23	4 years	Temporary documents	29 May 2020
Blanca	Female	Western Europe	35	1 year	Permanent documents	17 June 2020
Linh	Female	South-East Asia	30	2 years	Temporary documents	3 June 2020
Irene	Female	Eastern Europe	37	1 year	Temporary documents	14 June 2020
Lucia	Female	South America	38	6 years	Temporary documents	12 June 2020
Bogdan	Male	Eastern Europe	38	3 years	Temporary documents	10 June 2020
Bussaba	Female	South-East Asia	44	2 years, 6 months	Temporary documents	1 June 2020
Lamai	Female	South-East Asia	38	4 years	Temporary documents	2 June 2020
Sarah	Female	North America	36	2 years	Temporary documents	17 June 2020
Thomas	Male	Southern Africa	38	3 years	Temporary documents	3 June 2020

^1^ If you have submitted an application for a residence permit legally, you can have a procedural stay. This means that you can legally stay in Denmark when the application is being processed (www.nyidanmark.dk, 19 August 2021).

**Table 2 ijerph-18-08932-t002:** Thematic network analysis.

Basic Themes	Organizing Themes
- Finding out what is going on is difficult	1. I just want some clear answers
- Most information about COVID-19 is in Danish	
- Feeling left out of the response	
- Relying on help with translation of COVID-19-related information	2. Seeking to ‘be in the know’
- Facebook groups provide translation of Danish news	
- Reading news from home country to stay informed	
- Workplace provides information and support	

## Data Availability

The data are not publicly available. Requests to access the data used in this study can be sent to the corresponding author.

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
