# Peer review of "“I Just Want Some Clear Answers”: Challenges and Tactics Adopted by Migrants in Denmark When Accessing Health Risk Information about COVID-19"

_ijerph, 2021, doi:10.3390/ijerph18178932_

Round 1

Reviewer 1 Report

The authors have addressed my concerns and questions well. I don't have any other comments.

Author Response

Thank you very much for your kind feedback. 

Sincerely, 

Morten Skovdal & Rasmus Brønholt

Reviewer 2 Report

This qualitative research study and manuscript focus on important and less examined public health and communication challenges in COVID-19 prevention efforts, which is reaching migrants with information. As such, this study and manuscript provide timely and instructive information and insights into the challenges that migrants face when it comes to receiving and accessing needed information in a public health emergency as well as recommendations for communicating to migrants. The strengthens of this manuscript include its timeliness, relevance and value to public health communication and outreach efforts, and the insights that are surfaced. It is generally well organized and written, and lines 146-158 provide a strong and well-stated rationale for studies such as this one. 

My primary suggestions for strengthening the manuscript are:

  • I'm not sure the phrase "Lost in translation," which is attention getting, is an effective or helpful phrase, including in the title, given the study makes clear that a key shortcoming of the public health communication efforts was "little or no translation" of public health COVID-19-related information. "Lost in translation" implies or suggests a major problem was with how information was translated from Dutch into other languages. This study makes clear a major problem is the lack of translation of public health information into multiple other languages. In addition, the use of the question in the title is not well connected with what comes after. The "Challenges and tactics adopted by migrants in Denmark when accessing health risk information about COVID-19" put the focus in a different place. Thus, a more accurate characterization of the study, including in the title, would be something along the lines of "Where is the translation?" or "Lack of translation."
  • There are a few places where the wording or phrasing could be strengthened. These include page 1, lines 71-74, where "brings" would be a better word than "spurred" and "accurate" would be a better word than "correct"; page 1, line 78, where "suffer" should be "suffering"; page 1, line 80 where ". . . giving them an equal chance to find effective ways to protect themselves from infection. . ." would be better than "giving them an equal chance to creatively find ways of shielding themselves from infection"; and page 8, line 718, "utilized" would be a better word than "circumvented." 
  • Page 2, lines 159-170 - this paragraph would be further strengthened by noting that not addressing migrants' linguistic diversity does not just leave migrants underserved and vulnerable to infectious diseases like COVID-19, it also adversely impacts the broader community or population. When public health efforts fail to encompass all people living in a community or area, the people who are ignored not only are highly susceptible to infection and illness, their susceptibility enables transmission of the virus to others and can enable continued changes in the virus. 
  • Page 1, line 78 - reference is made to psychosocial consequences, but no examples are mentioned. It would help to mention examples.
  • In the illustrative participant narratives, I would recommend deleting the "uhs," "uhm," "ehm," etc. because they are distracting and have no value. 
  • Table 2 seems to be missing text in column 2. 
  • In the Discussion, lines 727-734, it is stated that "Migrants would had stayed in Denmark for several years and therefore were more established, appeared to face fewer challenges in navigating health risk information about COVID-19." This study's design, and the number of participants who lived in Denmark for several years (i.e., 4 people who lived in Denmark for three or more years), do not support such a strong assertion or generalization. It would be better to say that those who lived in Denmark longer MAY have face fewer challenges and to note this is something that would benefit from future research.
  • A few edit suggestions: page 1, line 71, "is" should be "are"; page 2, line 171, the word "either" should be dropped because multiple options are listed and the first use of the word "and" should be changed to "but" (i.e., . . . editorials, BUT we still have. . .); page 6, line 536, the sentence should start with "For example,"; page 7, 636, the would "that" is needed after "social media platforms"; and page 9, line 874, "groups" should be "individuals" (i.e., . . . resourcefulness of migrant individuals. . .).

Author Response

Dear reviewer

Thank you for your thorough review and positive evaluation of our manuscript.

We have revised the manuscript according to your feedback. Attached, you will find a point-by-point response to the questions and comments.

Changes are marked with tracked changes.

We believe that the manuscript has improved even more, and we are looking forward to receiving your feedback towards publication of the manuscript.

Best regards,

Rasmus Brønholt and Morten Skovdal

This manuscript is a resubmission of an earlier submission. The following is a list of the peer review reports and author responses from that submission.

Round 1

Reviewer 1 Report

Based on a thematic analysis of 11 interviews, this study explored how young migrants in Denmark face challenges and adopt tactics to access information about COVID-19 during the early stage of the pandemic. This project is timely and relevant, and the findings may have implications for future health and risk communication. I have several clarification questions and comments, which I elaborate on below.

  1. Page 2, Line 62: “The linguistic challenges associated with being a migrant in a time 61 of crisis have also been reported in other studies.” I’d suggest the authors elaborating on “other studies.” For instance, the authors may use a sentence or two to summarize the study contexts and main findings.

  1. Page 3, Line 92: It seems that some participants answered the survey in Danish. Did the authors collect information about their language background? How many participants were not fluent in Danish? Is English the first language for all participants? If the information is available, I’d suggest the authors adding each participant’s first/native language, as well as their fluency in English and Danish to Table 1. This will better help readers to interpret the theme of “lost in translation.”

  1. The authors recruited migrants of any age, but decided to focus on young participants only. Please justify the decision. In addition, please clarify if “who had either not been in the country for a very long time or who came by 97 themselves” (page 3, Line 97) is an inclusion criterion of this study. Why is this criterion important here?

  1. In a footnote of Table 1, please briefly explain what “procedural stay” means. While “temporary documents” and “permanent documents” seem to be self-explanatory, readers outside Demark may not be familiar with “procedural stay.”

  1. The interview was conducted in English, which may be the second language for some participants. How may this affect the interview and findings? I’d suggest the authors acknowledge and address this in the limitation section.

  1. I am wondering what the “29 codes” are. Can the authors provide this information in an appendix?

  1. Table 2: “Finding out what is going on when is difficult.” Do the authors mean “Finding out what is going on is difficult.”?

  1. Instead of calling the two higher-level themes “organizing themes,” I am wondering if “overarching theme” may be a better term?

  1. The authors did a great presenting their findings. Yet, I’d suggest the authors moving the “linguistic difficulties” section (Line 174-194) to the front. In this way, the text and the themes shown in Table 2 will parallel each other.

  1. I think the authors did a good job interpreting the findings. The discussion about digital tools is also valuable. However, I feel the “so-what” question remains unanswered. In other words, what are the practical implications for public health and government agencies? What can they and us (as researchers) learn from this study? I’d suggest the authors adding a brief section to address these questions.

  1. Please proofread the paper. Some words included an unnecessary dash (-). For instance, “out-breaks” (Line 13), “high-lights” (Line 22), avail-able (Line 48), cur-rently (Line 105), be-hind (Line 151), ad-dressed (Line 188), and “Re-public” (Line 281). The line space is also incorrect for the Conclusion session.

Reviewer 2 Report

The chosen topic is very interesting and the work could serve to give greater visibility to the problem of migrants when they are exposed to information or risk communication processes, as is happening in the pandemic caused by this coronavirus. I welcome the focus on this issue, especially in a country like Denmark. However, I believe that the chosen sample could be enriched and expanded, which would favor a greater scope, impact and validity of this work. The in-depth interview is an appropriate technique for this study, but I suggest that a greater number of people be interviewed, in order to show a more faithful picture of the problem and more conclusive results. With more data available, they could make use of some type of infographic or graph, which always favors the analysis and understanding of the phenomenon.

Reviewer 3 Report

Dear Authors,

The article presents contributions to the communication of health risks. Methods are adequately described and results are clearly presented. My recommendation is accept in present form.

Author Response

Reviewer 3:

Dear Authors,

The article presents contributions to the communication of health risks. Methods are adequately described and results are clearly presented. My recommendation is accept in present form.